# Pharmacogenetic strategies to mitigate cisplatin-induced ototoxicity in head and neck cancer: A cost-minimization analysis with the use of *GSTP1* c.313A>G genotyping

Ligia Traldi Macedo[1][*], Vinicius Eduardo Ferrari[2], Ericka Francislaine Dias Costa[3], Gustavo Jacob Lourenço[3], Luciane Calonga[4], Arthur Menino Castilho[4], Carlos Takahiro Chone[4], Carmen Silvia Passos Lima[1,3]

**1** Department of Oncology and Radiology, Faculty of Medical Sciences, University of Campinas (UNICAMP), Brazil, **2** School of Economics and Business (EcoN), Pontifical Catholic University of Campinas (PUCC), Campinas, Brazil, **3** Laboratory of Cancer Genetics (LAGECA), Faculty of Medical Sciences, University of Campinas (UNICAMP), Brazil, **4** Department of Ophthalmology and Otorhinolaryngology, Faculty of Medical Sciences, University of Campinas, Brazil

☉ These authors contributed equally to this work.

* ligia.macedo@alumni.harvard.edu

## Abstract

### Background

Cisplatin is a cornerstone agent in the treatment of head and neck squamous cell carcinoma. Nonetheless, cisplatin is associated with significant ototoxicity, leading to substantial irreversible hearing loss. Pharmacogenetic testing offers a potential strategy to identify patients at risk, allowing for personalized treatment plans that could mitigate this adverse event and reduce related costs.

### Methods

This study aimed to evaluate the economic impact of pharmacogenomic screening. A cost-minimization analysis was conducted using a decision-analytic model incorporating Bayesian inference through Metropolis-Hastings algorithm. Probabilities of ototoxicity were derived from existing literature. The costs of standard treatment were compared with those of a genotype-guided approach, assessing potential cost savings related to the need for audiological interventions.

### Results

In 250 patients, the genotype-guided approach reduced the incidence of moderate-to-severe ototoxicity from 29% to 18% among low-risk patients while avoiding cisplatin in high-risk individuals. The total savings over 10 years were estimated at US$13,077.73 (Credible Interval US$11,026.07 to US$14,147.61), driven primarily by reduced costs for audiological interventions. The break-even point for

**Data availability statement:** All relevant data are within the paper and its Supporting Information files.

**Funding:** The author(s) received no specific funding for this work.

**Competing interests:** The authors have declared that no competing interests exist.

cost-effectiveness was achieved when at least 275 patients were tested annually. Sensitivity analyses demonstrated that the cost savings remained robust under variations in patient volume and testing costs.

## Conclusions

Implementing pharmacogenomic screening in the management of head and neck squamous cell carcinoma patients treated with cisplatin may offer significant economic benefits. Personalized treatment plans based on genetic risk for ototoxicity could potentially not only enhance patient-related outcomes but also contribute to more efficient use of healthcare resources. Prospective, randomized evaluations would be ideal to confirm these findings.

## Introduction

Head and neck squamous cell carcinoma (HNSCC) comprises a group of neoplasms arising from the mucosal epithelium lining critical anatomical sites, including the larynx, pharynx, oral cavity, as well as nasopharynx. It is a highly incident malignancy, often linked to tobacco use, alcohol consumption, and human papillomavirus (HPV) infection [1]. The untreated trajectory of SCCHN precipitates severe complications, encompassing tumor-related hemorrhage, malnutrition due to impaired oral intake, airway obstruction, secondary infections and electrolyte imbalances such as paraneoplastic hypercalcemia [2]. Representing over 90% of head and neck tumors [3], HNSCC is among the most common globally, with nearly 900,000 new cases and over 440,000 deaths estimated annually [4]. About 75% of patients present with locally advanced disease upon diagnosis [5], to which cisplatin (CDDP)-based chemoradiotherapy is usually administered either as a definitive or adjuvant approach [6]. CDDP is considered the standard of care for its reported improvement in disease-related outcomes, with complete response rates ranging from 55 to 71% [7,8] and absolute gain in overall survival of 6.5% in 5 years [6].

Despite its clinical efficacy, CDDP is often limited by dose-dependent toxicities, including nephrotoxicity, neurotoxicity, and ototoxicity [9–11]. Among these, ototoxicity is of particular concern due to the risk of permanent bilateral hearing loss and profound impact on patients' quality of life, with the severity often progressing with higher cumulative doses [12]. Approximately 20–40% of CDDP-treated patients develop significant hearing loss, depending on age, treatment regimen, and genetic predispositions [13–15]. High-frequency hearing loss is often the earliest symptom, which progresses to affect speech frequencies. CDDP-induced ototoxicity primarily derives from oxidative stress and cochlear inflammation, leading to damage to the hair cells of the organ of Corti and auditory neurons [16,17]. This condition can severely impair communication, contributing to social isolation, depression, and cognitive decline, particularly in older adults [18–20]. Despite its prevalence and burden, there are no effective otoprotective agents approved for clinical use to prevent CDDP-induced ototoxicity to this date [15,21]. Current strategies focus on early detection through

audiometric monitoring and adjusting treatment regimens for at-risk patients [22]. Potentially less ototoxic alternative agents such as cetuximab [23] or carboplatin [24] have failed to demonstrate non-inferiority to CDDP regarding efficacy. Docetaxel, an anti-microtubule agent, has also been studied in patients deemed unfit for CDDP, with positive results in overall survival [25]. Direct comparisons to CDDP are yet to be performed in the overall population.

Moreover, the management of ototoxicity, irrespective of its underlying cause, imposes significant economic burden due to the requirement of hearing aids and long-term follow-up care. It is estimated that 403.3 million people suffer from moderate to severe hearing loss globally, necessitating the use of hearing aids or cochlear implants [26]. The estimated annual economic impact consists of approximately US$ 750 billion, taking into account multidisciplinary support, acquisition and maintenance of auditory devices, and years lived with disability [27]. Specifically in Brazil, the national Unified Health System (SUS) spent approximately R$ 89,437,570.86 on audiology services between 2009 and 2018, with an annual expenditure of R$ 9 million [28]. Untreated hearing loss can also lead to the development of comorbidities, further increasing medical expenses for indirect conditions [18,19]. Additionally, hearing loss often results in decreased productivity or even workforce marginalization, negatively affecting individual income and imposing additional societal costs through the need for social assistance programs and work rehabilitation [29].

The study of genetic factors influencing drug response known as pharmacogenomics has emerged as a promising field for predicting safety and efficacy of chemotherapy [30]. Genetic variants involved in drug metabolism, detoxification, and DNA repair pathways have been implicated in CDDP-induced ototoxicity [31]. Among these, the glutathione transferase P1 *GSTP1* c.313A>G variant, which encodes a key enzyme in glutathione-mediated detoxification [32], has been identified as a significant predictor of ototoxicity risk [33–35]. The detoxification of CDDP, as such, occurs predominantly through glutathione-transferases, enzymes catalyzing the conjugation of toxic substrates to glutathione molecules, rendering them water-soluble for cellular diffusion and renal excretion [32]. Patients with the *GSTP1* c.313 AG or GG genotypes have reduced glutathione detoxification activity, leading to CDDP intracellular accumulation, prolonged cochlear exposure to CDDP and increased oxidative damage [17,36]. These genetic variants were associated with a 4.3-fold higher risk of moderate-to-severe ototoxicity compared to individuals with the AA genotype [33]. Other genetic markers have also been linked to CDDP-related toxicities, highlighting the potential for a multifactorial risk assessment model [37–39].

Considering the global prevalence of hearing impairment and the economic consequences of ototoxicity induced by CDDP, a comprehensive exploration of the financial implications and cost-effective strategies for identifying high-risk patients is imperative. Pharmacogenetics holds promise for financially optimizing treatment outcomes. In resource-constrained settings, where there is limited access to advanced medical interventions, the application of pharmacogenetic principles could potentially lead to more cost-effective healthcare. This approach may reduce the financial burden on healthcare systems by avoiding unnecessary treatments, optimizing drug choices, and preventing adverse reactions, ultimately contributing to improved patient outcomes and more sustainable healthcare practices. By pinpointing genetic variants as *GSTP1* c.313A>G, tailored therapeutic strategies can be implemented. The integration of pharmacogenomic testing into clinical practice offers an opportunity to identify high-risk individuals and individualize treatment accordingly. By substituting CDDP with less ototoxic alternatives, such as docetaxel [25], or implementing modified dosing strategies [40], pharmacogenomic-guided therapy could reduce the incidence of hearing loss and its associated costs.

Cost-effectiveness studies have demonstrated potential economic and clinical benefits of genetic screening for adverse drug reactions in oncology, such as fluoropyrimidine-related toxicities [41] and irinotecan-induced neutropenia [42]. Nervertheless, there is limited literature on the cost-effectiveness of pharmacogenomic testing specifically for CDDP-induced ototoxicity. A single study validated genotyping as a cost-effective strategy for ototoxicity risk selection with focus on catechol-O-methyltransferase (*COMT*) and thiopurine S-methyltransferase (*TPMT*) encoding variants [43]. However, it was primarily directed to the pediatric population of patients, where HNSCC are not prevalent. Studies regarding adult HNSCC patients under chemoradiation are yet lacking. Hence, the primary aim of this study was to evaluate the financial impact of *GSTP1* c.313A>G testing in HNSCC patients who are candidates for definitive chemoradiotherapy, specifically

assessing the financial benefits of selectively prescribing docetaxel to individuals with the high-risk *GSTP1* c.313 G allele, analyzing potential cost savings through reduced incidence of moderate-to-severe ototoxicity and decreased need for hearing aids and related support, following risk stratification and individualized treatment plan.

## Materials and methods

### Decision analytic model

This study comprises a cost-minimization analysis incorporating *GSTP1* c.313A>G genetic testing into the management of HNSCC patients treated with CDDP-based chemoradiotherapy. The analysis compared two approaches: the standard of care, where all patients received CDDP without prior genotyping stratification, and a genotype-guided therapy selection, where patients with *GSTP1* c.313 AG or GG were identified as unfit to CDDP and submitted to docetaxel as a radiosensitizer to reduce the risk of ototoxicity. The study was conducted under a healthcare payer perspective and applied a 10-year time horizon to evaluate both direct and long-term costs.

To model transitional probabilities and costs, a decision-analytic tree was designed (Fig 1). Patients in the genotype-guided approach underwent genetic testing before treatment through real-time polymerase chain reaction (PCR), aiming to detect *GSTP1* c.313A>G variants and direct tailored therapy. Regarding chemotherapy, the standard

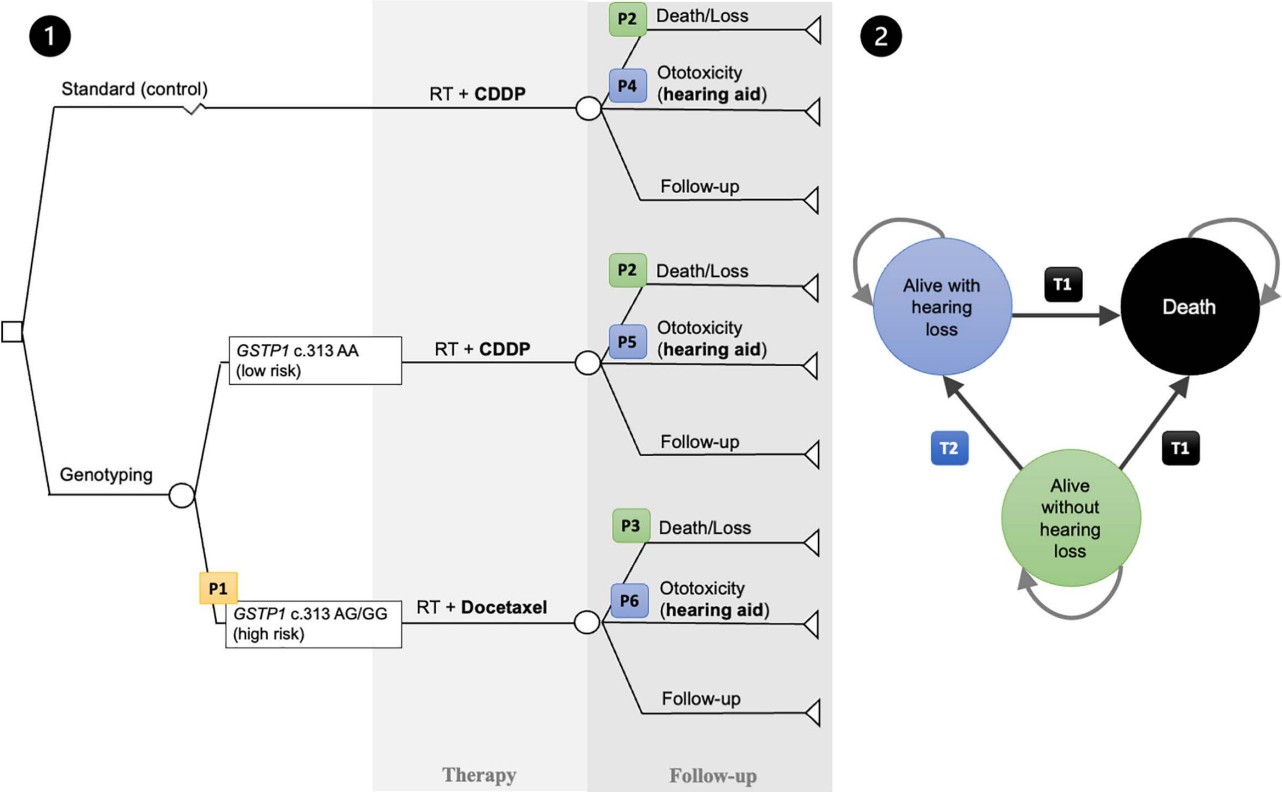

**Fig 1. Decision-analytic tree (1) and Markov transitional states (2).** P1: probability of high-risk group for moderate to severe ototoxicity (*GSTP1* c.313 AG/GG genotypes); P2: probability of death or loss to follow-up during therapy in cisplatin-treated patients; P3: probability of death or loss to follow-up during therapy in docetaxel-treated patients; P4: probability of moderate to severe ototoxicity in the conventional arm; P5: probability of moderate to severe ototoxicity in low-risk patients treated with cisplatin; P6: probability of moderate to severe ototoxicity in high-risk patients treated with docetaxel; T1: transitional probability for death; T2: transitional probability for hearing loss in asymptomatic individuals. A: adenine; CDDP: cisplatin; G: guanine; *GSTP1*: glutathione transferase P1 gene; RT: radiotherapy.

care arm and low-risk patients (AA genotype) in the genotype-guided arm received CDDP at a dose of 100 mg per square meter of body surface area in three-weekly intervals, totaling 3 cycles. In the high-risk genotype-guided arm (AG and GG genotypes), patients were submitted to docetaxel at a dose of 15 mg per square meter of body surface area weekly for seven weeks. Supportive care, including hydration, antiemetics, and monitoring for nephrotoxicity and ototoxicity, constituted usual practice. Radiotherapy regimen and delivery dose were similar in both groups. The indication for hearing aids and associated support was the presence of moderate-to-severe ototoxicity, defined as National Cancer Institute Criteria for Adverse Events (NCI CTCAE 4.03) [44] hearing loss grade 3 or greater, which consists of the most widely adopted classification of toxicity in cancer patients under treatment, with sufficient literature data. Based on the recommendation of hearing aids for sensorineural hearing loss starting at 35–40 dB (decibel) thresholds [45], it can be inferred that patients with NCI CTCAE grade 3 or higher ototoxicity fall into this category, as hearing losses exceeding 25 dB across more than three consecutive frequencies result in thresholds above this range. The following transitional probabilities were retrieved for incorporation into the model:

- The probability of high-risk for moderate to severe ototoxicity *GSTP1* c.313 AG or GG genotypes (P1);

- The probabilities of treatment discontinuation or death with CDDP (P2);

- The probabilities of treatment discontinuation or death with docetaxel (P3);

- The risk of moderate-to-severe ototoxicity in patients undergoing conventional treatment (radiotherapy and CDDP) without genotyping (P4);

- The risk of moderate-to-severe ototoxicity in low-risk patients (*GSTP1* c.313 AA genotype) treated with CDDP (P5);

- The risk of moderate-to-severe ototoxicity in high-risk patients (*GSTP1* c.313 AG or GG genotypes) treated with docetaxel (P6);

- The annual risk of death during follow-up (T1);

- The annual risk of hearing loss in previously asymptomatic patients during follow-up (T2).

Bayesian inference, using the Metropolis-Hastings algorithm [46], was used to estimate these probabilities based on prior data and observed outcomes. For the Markov model analysis [47], as depicted in Fig 1, transitional probabilities between different states (in this case, alive with hearing loss, alive without hearing loss, and death) were obtained from the literature, considering the cumulative risk over ten years. In the case of mortality (T1), a median follow-up mortality rate of 77% over 9.2 years was considered based on a meta-analysis for this group [48]. For hearing loss in healthy individuals (T2), a 4.2% rate over ten years, adjusted for age, according to Global Burden of Disease (GBD) estimates, was used [49].

### Cost incorporation

Testing costs were calculated based on laboratory expenditures, including reagent prices, equipment amortization, and personnel time, as detailed in Table 1. To better reflect clinical practice, we conducted comprehensive cost calculations, taking into account both single-test scenarios and batch sample processing, accounting for the use of controls and 10% of potential losses (cost inputs for DNA extraction, real-time PCR, equipment, manpower and value per sample are detailed in S1, S2, S3, S4 and S5 Tables in Supporting Information, respectively). The selected real-time PCR equipment can process up to 96 tests per cycle, while DNA extraction allows for the simultaneous handling of 12 samples. Simulations were performed for batch sizes of up to five simultaneous samples, and the resulting cost savings were analyzed.

The initial investment included the acquisition of equipment necessary for *GSTP1* c.313A>G testing via real-time PCR (StepOnePlus Real-Time PCR System and Software, Thermo Fisher Scientific Inc.), with an annual amortization rate of

**Table 1. Summarized data on cost inputs.**

| Recurrent costs | Overall cost (US$) |
| --- | --- |
| *Chemotherapy and supportive agents/material* | |
| Cisplatin 100 mg/m² every 21 days (3 cycles) | 307.02 |
| Docetaxel 15 mg/m² weekly (7 weeks) | 270.29 |
| *Genotyping (with real-time PCR, considering material and manpower)* | |
| 3 simultaneous samples | 26.27 |
| 5 simultaneous samples | 17.12 |
| *Use of hearing aids in patients with moderate to severe ototoxicity (material and support)* | |
| Annual cost per patient (1 device) | 73.81 |
| **Investment and amortization** | **Overall cost (US$)** |
| StepOne Plus PCR System | 38,340.00 |
| Annual amortization of 20% | 7,668.00 |

mg: milligrams; m²: square meter (body surface area); PCR: polymerase chain reaction; US$: United States Dollars.

20% [50]. While the aforementioned equipment can also be utilized for unrelated real-time PCR assays, the full cost was factored into the amortization calculations. As a result, the estimated annual amortization cost may be slightly inflated. Nevertheless, this conservative approach was chosen to account for the uncertainty surrounding additional procedures and potential cost savings.

The costs of therapy included drug acquisition, administration, and supportive care components. Drug prices were sourced from local hospital suppliers, while genetic testing costs were derived from laboratory protocols (cost inputs for chemotherapy are further detailed in S6 Table, Supporting Information). Costs associated with ototoxicity management were derived from the High Complexity Procedures Authorization table for Brazil's Unified Health System (SUS) and included the use of hearing aids, audiological evaluations, and device maintenance (available in the Virtual Health Library of the Ministry of Health) [28,51]. Patients with moderate-to-severe ototoxicity were assumed to require long-term care and hearing aids as suggested in literature [45], with follow-up services extending over a 10-year period (S7 Table, Supporting Information).

Inputs for the analysis regarding local treatment, ototoxicity management and manpower were obtained in local currency (Brazilian reais – BRL) and converted to US Dollars (US$), based on the expected median exchange rate of 1 US$ for 5.50 BRL (Focus-BC report for 24 November 2024, from the Central Bank of Brazil). The financial inputs on laboratory reagents, tubes and machinery were originally imported, obtained in US$.

For future costs in the Markov model, an annual discount rate aligned with the estimates provided by the Brazilian Association of Financial and Capital Market Entities (ANBIMA) was applied. This rate corresponds to the par real yield of a 10-year Brazilian Inflation-Protected Bond, which was 6.79% as of December 11, 2024 [52].

## Statistical analyses

The cost-minimization analysis calculated total costs for each treatment arm by the sum of expenditures for chemotherapy, genetic testing, and ototoxicity management. The difference in total costs between the two arms represented the potential savings associated with implementing genetic testing. In the standard care arm, costs included chemotherapy-related expenses and long-term audiological care. In the genotype-guided arm, costs included genetic testing for the entire cohort, CDDP-related costs for low-risk individuals, docetaxel-related costs for high-risk patients, and audiological

care. A Bayesian Bernoulli model was employed, where posterior probability distributions were derived from previously published data (priors) [53–57] and binomial likelihood functions reflecting the local data previously published [31], except for the probabilities involved in docetaxel treatment [25,58,59].

Once the probabilistic model was constructed, Markov chain Monte Carlo simulations with 12,500 iterations were performed using the Metropolis-Hastings algorithm, with an initial burn-in phase of 2,500 iterations. The Metropolis-Hastings algorithm repeatedly sampled random values generated by the Bayesian Bernoulli model to estimate key parameters, including medians and credible intervals (CIs) for the posterior probabilities. These medians were then assigned to the corresponding hypothetical branches to calculate the total costs per arm (detailed calculations in Supplementary Material, section 1). Quality control for the simulations included the assessment of trace plots, histograms, efficiency metrics, and autocorrelation graphs. All computations were performed using Stata/IC 15 (StataCorp LLC 2017, College Station, TX).

## Ethics statement

This study presents a cost-minimization analysis that integrates genetic testing for *GSTP1* c.313A > G into the management of patients with HNSCC undergoing CDDP-based chemoradiotherapy. A decision-analytic model is employed to compare the costs of standard treatment versus a genotype-guided approach. All data for estimating cost parameters and calculating model transition probabilities were obtained from existing literature (as shown in Table 2). Additionally, probabilities for the annual risk of death during follow-up [48] and hearing loss [49] were sourced from the literature. Since our Bayesian Bernoulli model does not involve direct research on human subjects or the use of identifiable data, this study did not require prior approval from a Research Ethics Committee.

## Results

### Bayesian inferential probabilities and distribution

Based on the presented data and the most relevant findings regarding the risk of moderate-to-severe ototoxicity, the *GSTP1* c.313A > G variant was selected for the cost-reduction simulation due to the robustness of its data in the literature. Probabilities derived using the Markov Chain Monte Carlo simulations are detailed in Table 2, including medians and CIs. These probabilities formed the basis for calculating costs in the first year as cumulative costs over ten years. To estimate the probability of moderate-to-severe ototoxicity in low-risk genotype patients (*GSTP1* c.313 AA), results from a non-informative prior distribution were preferred. This decision was due to the similarity of results when compared to data from a cohort of pediatric patients with different tumor types and treatment.

Table 2. Markov Chain Monte Carlo simulations.

| Probabilities | Observed data (Binomial) | | | Prior data (Beta) | | | Simulation results | |
|---|---|---|---|---|---|---|---|---|
| | Events | No events | Ref. | Events | No events | Ref. | Median | CI |
| *GSTP1* c.313 AG or GG – High Risk (P1) | 49 | 40 | [31] | 178 | 114 | [55] | 0.42 | 0.37 - 0.47 |
| Death or loss to follow-up in CDDP (P2) | 3 | 89 | [31] | 8 | 166 | [53] | 0.03 | 0.02 - 0.06 |
| Death or loss to follow-up in docetaxel (P3) | 2 | 89 | [58] | 4 | 176 | [25] | 0.02 | 0.008 - 0.04 |
| Overall moderate-to-severe ototoxicity (P4) | 26 | 63 | [31] | 58 | 91 | [57] | 0.35 | 0.29 - 0.41 |
| Moderate-to-severe ototoxicity in low-risk CDDP (P5)* | 7 | 33 | [31] | NI | NI | NA | 0.18 | 0.08 - 0.31 |
| Moderate-to-severe ototoxicity in low-risk CDDP (P5)* | 7 | 33 | [31] | 4 | 20 | [56] | 0.17 | 0.09 - 0.27 |
| Moderate-to-severe ototoxicity in docetaxel (P6) | 1 | 16 | [54] | 10 | 91 | [60] | 0.09 | 0.04 - 0.15 |

CDDP: cisplatin; CI: credibility interval; *GSTP1*: glutathione transferase P1 encoding gene; NA: not applicable; NI: non informative; Ref.: reference;
* Moderate-to-severe ototoxicity in low-risk was assessed on either data from pediatric cancer population or as a non-informative prior.

## Costs simulations in the first year

For the first year of follow-up, the number of samples tested simultaneously for PCR significantly influenced total costs. Testing individual samples resulted in a financial deficit for the genotype-guided arm, whereas two simultaneous samples yielded positive savings for cohorts of 450 patients or more. S8 Table (Supporting Information) illustrates the cost reduction across different numbers of simultaneous tests and total patients analyzed. These findings demonstrate a progressive reduction in the number of patients required for cost savings as the number of samples processed simultaneously increased. For example, analyzing five samples simultaneously in a cumulative number of 500 patients resulted in total savings of US$11,578.55. The threshold for cost minimization also varied, ranging from 275 patients in the three-sample analysis to 200 patients in the five-sample simultaneous testing, as depicted in Fig 2.

The median cost per patient in the conventional arm was US$368.19 (CI US$359.20 to US$371.04). In the genotyping arm, total costs per patient varied depending on the number of samples processed simultaneously and cumulative patient cohort (**Table 3**). When testing three samples, the median cost reduction was US$14.01 (CI US$7.31 to US$15.72), resulting in a total expenditure of US$354.18 per patient for a cohort of 500 patients. For five samples, potential savings increased to US$23.16 (CI US$16.46 to US$24.87), with a total cost of US$345.03 per patient (S9 Table, Supporting Information).

## Cost simulations over ten years of follow-up

After adjusting for device amortization rates, annual discount rates, and risks of mortality and hearing loss in asymptomatic individuals, consolidated data over a ten-year follow-up period revealed total cumulative savings of US$13,077.73 (CI US$11,026.07 to US$14,147.61) with genotype-guided therapy, considering a cohort of 250 patients. These savings comprise US$10,540.71 in reduced spending on hearing aids (CI US$11,403.04 to US$8,887.07) and US$2,537.02 on audiological support services including regular audiometry assessment and clinical appointments (CI US$2,744.57 to US$2,139.01). Table 4 summarizes the annual and cumulative savings over the ten-year period, while the detailed results, including the number of active patients at different Markov states, medians, inferior and superior limits, are provided in S10, S11 and S12 Tables in Supporting Information.

Maximum annual reductions in cost were observed in the third year of follow-up, where annual savings per patient were US$12.44 (CI US$10.49 to US$13.45), for the three-sample simultaneous testing in 250 patient's scenario. This figure gradually decreased to an annual reduction of US$7.62 (CI US$6.43 to US$8.25), reflecting the progressive reduction in active patient numbers over time due to mortality.

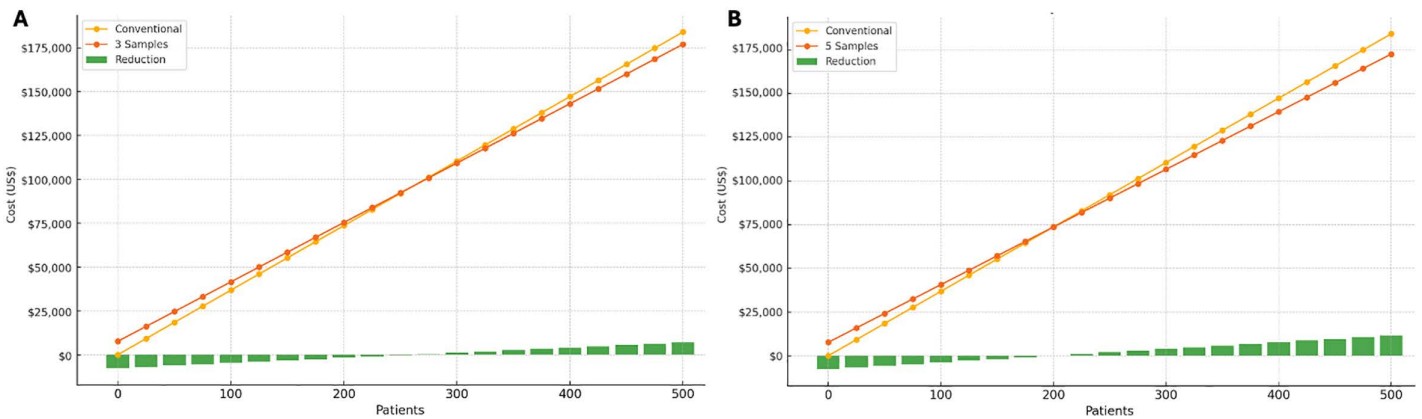

**Fig 2. Graphical representation of the total costs in the first year for the conventional arm and _GSTP1_ c.313A > G genotyping and the cost reduction, based on the number of simultaneous samples (3 samples in A, 5 samples in B) and the cumulative volume of analyzed patients.** US$: United States Dollars.

**Table 3. Total costs per patient in the first year for the *GSTP1* c.313A>G genotyping group and the differences compared to the conventional arm, considering simultaneous genotyping of three samples.**

| Patients | Cost (Median) | Credibility Interval | Difference (Median) | Credibility Interval |
|---|---|---|---|---|
| 25 | US$645.56 | US$634.86 to US$655.11 | -US$277.37 | -US$284.07 to -US$275.66 |
| 50 | US$492.20 | US$481.50 to US$501.75 | -US$124.01 | -US$130.71 to -US$122.30 |
| 75 | US$441.08 | US$430.38 to US$450.63 | -US$72.89 | -US$79.59 to -US$71.18 |
| 100 | US$415.52 | US$404.82 to US$425.07 | -US$47.33 | -US$54.03 to -US$45.62 |
| 125 | US$400.19 | US$389.48 to US$409.74 | -US$32.00 | -US$38.69 to -US$30.29 |
| 150 | US$389.96 | US$379.26 to US$399.51 | -US$21.77 | -US$28.47 to -US$20.06 |
| 175 | US$382.66 | US$371.96 to US$392.21 | -US$14.47 | -US$21.17 to -US$12.76 |
| 200 | US$377.18 | US$366.48 to US$386.73 | -US$8.99 | -US$15.69 to -US$7.28 |
| 225 | US$372.92 | US$362.22 to US$382.47 | -US$4.73 | -US$11.43 to -US$3.02 |
| 250 | US$369.51 | US$358.81 to US$379.07 | -US$1.32 | -US$8.02 to US$0.38 |
| 275 | US$366.73 | US$356.02 to US$376.28 | US$1.46 | -US$5.23 to US$3.17 |
| 300 | US$364.40 | US$353.70 to US$373.95 | US$3.79 | -US$2.91 to US$5.50 |
| 325 | US$362.44 | US$351.73 to US$371.99 | US$5.75 | -US$0.94 to US$7.46 |
| 350 | US$360.75 | US$350.05 to US$370.30 | US$7.44 | US$0.74 to US$9.15 |
| 375 | US$359.29 | US$348.59 to US$368.84 | US$8.90 | US$2.20 to US$10.61 |
| 400 | US$358.01 | US$347.31 to US$367.56 | US$10.18 | US$3.48 to US$11.89 |
| 425 | US$356.89 | US$346.18 to US$366.44 | US$11.31 | US$4.61 to US$13.01 |
| 450 | US$355.88 | US$345.18 to US$365.43 | US$12.31 | US$5.61 to US$14.02 |
| 475 | US$354.99 | US$344.28 to US$364.54 | US$13.20 | US$6.51 to US$14.91 |
| 500 | US$354.18 | US$343.48 to US$363.73 | US$14.01 | US$7.31 to US$15.72 |

US$: United States Dollars.

**Table 4. Cost reduction associated with the use of *GSTP1* c.313A>G genotyping over a ten-year follow-up period, considering genotyping with three samples in a population of 250 patients.**

| Year | Patients | Annual Reduction | Credibility Interval | Average Annual Reduction per Patient | Credibility Interval |
|---|---|---|---|---|---|
| 1 | 231 | US$442.93 | US$373.44 to US$479.16 | US$1.92 | US$1.62 to US$2.08 |
| 2 | 213 | US$370.85 | US$312.67 to US$401.19 | US$1.74 | US$1.47 to US$1.88 |
| 3 | 197 | US$2,444.75 | US$2,061.22 to US$2,644.76 | US$12.44 | US$10.49 to US$13.45 |
| 4 | 181 | US$2,104.16 | US$1,774.05 to US$2,276.30 | US$11.60 | US$9.78 to US$12.55 |
| 5 | 167 | US$1,811.01 | US$1,526.90 to US$1,959.17 | US$10.81 | US$9.12 to US$11.70 |
| 6 | 155 | US$1,558.71 | US$1,314.17 to US$1,686.22 | US$10.08 | US$8.50 to US$10.91 |
| 7 | 143 | US$1,341.55 | US$1,131.09 to US$1,451.30 | US$9.40 | US$7.93 to US$10.17 |
| 8 | 132 | US$1,154.65 | US$973.51 to US$1,249.11 | US$8.77 | US$7.39 to US$9.49 |
| 9 | 122 | US$993.79 | US$837.88 to US$1,075.09 | US$8.18 | US$6.89 to US$8.84 |
| 10 | 112 | US$855.34 | US$721.15 to US$925.31 | US$7.62 | US$6.43 to US$8.25 |
| Total | – | **US$13,077.73** | **US$11,026.07 to US$14,147.61** | – | – |

US$: United States Dollars.

## Discussion

Based on the presented findings, it was possible to infer that *GSTP1* c.313A>G genotyping could emerge as a promising tool for identifying the risk of moderate-to-severe ototoxicity in HNSCC patients treated with CDDP-based chemoradiotherapy. Additionally, the availability of prior literature data, enabling the application of the Metropolis-Hastings algorithm and

sensitivity analyses, supported its selection for this simulation. The first-year follow-up analysis demonstrated that genotyping could lead to significant cost savings if performed simultaneously on a larger number of samples and with a higher volume of patients (from 275 patients onwards, for three simultaneous samples tested).

Although no direct comparison exists for these specific data, cost reductions associated with genotyping other variants (e.g., *TPMT* and *COMT*) for risk stratification have also been shown to be beneficial. A study by Dionne and colleagues [43] conducted in pediatric patients used similar criteria for moderate-to-severe ototoxicity and reported comparable cost-saving outcomes. Furthermore, the use of combined genotypes with other potential genetic predictors could enhance cost-effectiveness by offering more robust risk stratification and facilitating larger-scale simultaneous testing. These findings reinforce the relevance of genotyping as a potential strategy for addressing treatment-related toxicities. Nonetheless, due to the absence of comparative data in the literature and the limited sample size, this analysis focused solely on *GSTP1* c.313A > G.

The graphical representation of total first-year costs revealed a consistent trend of cost reductions with an increasing number of simultaneous PCR tests, highlighting the economic efficiency of the proposed strategy. Extending the analysis to a 10-year follow-up period, which incorporated mortality rates and hearing loss progression in asymptomatic individuals, further demonstrated that *GSTP1* c.313A > G genotyping is not only clinically effective but also financially viable. The cumulative savings of US$13,077.73 over this interval, considering a population of 250 patients, confirms the contribution of this approach to reducing costs associated with hearing aid and support services. It is worth noting that cost reductions could be even more pronounced with standardized controls for PCR and optimized human resource allocation. The calculations also accounted for a minimum of one device per patient, although some patients may require bilateral use of hearing aids. However, to ensure study clarity, additional parallel laboratory tasks affecting pricing as well as the use of two hearing devices were not considered. Despite this, the findings strongly support the cost-reduction potential of genotype-based treatment selection.

The findings of this study have strong potential for generalization across diverse healthcare settings. The use of *GSTP1* c.313A > G genotyping as a pharmacogenetic strategy to guide CDDP-based treatment decisions is an adaptable approach that can potentially benefit patients worldwide. Although this study is grounded in the context of the Brazilian healthcare system, the methodology and outcomes are in line with universal goals of reducing treatment-related toxicities and optimizing healthcare resources. The cost-minimization framework and risk stratification model can be tailored to distinct healthcare systems by adjusting local cost inputs and patient population characteristics. Additionally, the growing accessibility of genetic testing in different regions makes this strategy increasingly feasible, even in resource-limited settings. These analyses further enforce the value of personalized medicine and the potential of pharmacogenetic approaches in oncology to improve patient outcomes and enhance healthcare efficiency on a global scale.

This study, however, relies on several critical assumptions that merit discussion. Firstly, it was assumed that the efficacy of docetaxel-based regimens [25,58] is comparable to CDDP in the treatment of locally advanced HNSCC chemoradiotherapy. While CDDP remains the agent of choice, emerging evidence supports the use of alternative therapies [23–25] for patients who are considered ineligible due to other comorbidities or risk factors [61], taking into account the improvement in survival outcomes in comparison to radiotherapy alone. However, direct comparisons of long-term outcomes between CDDP and docetaxel for the overall HNSCC population in randomized trials are yet to be performed. This assumption introduces some uncertainty into the analysis, highlighting the need for further clinical studies to validate the equivalence of these treatments. Docetaxel was the agent of choice in this simulation, since the remaining treatment alternatives available to CDDP (carboplatin and cetuximab) have failed to demonstrate non-inferiority [24,62]. Furthermore, the study assumed that the auditory impact of docetaxel, when in combination with radiotherapy, is similar to that of radiotherapy alone. This assumption was necessary due to the lack of published audiometric data on docetaxel-treated patients in this specific setting. Although docetaxel-induced ototoxicity is rare, the absence of controlled data for this adverse event warrants further investigation. Additionally, the model relied on Bayesian probabilities derived from literature priors. For

low-risk genotypes, non-informative priors were used, as the available evidence was limited to pediatric populations with distinct tumor types and treatments [56]. While Bayesian methods allow for robust estimation under uncertainty, the generalizability of these probabilities to broader HNSCC populations should be validated.

Additionally, heterogeneity in patient populations, including presumed geographic variation in *GSTP1* variant prevalence, age-related susceptibility to ototoxicity, and comorbidities, may influence the generalizability of results. The model's application to a diverse cohort highlights its adaptability, but further subgroup analyses could elucidate differences in cost-effectiveness across varying clinical scenarios. Prospective clinical trials incorporating these diverse subpopulations would enhance confidence in the findings and provide a more nuanced understanding of the benefits and limitations of implementing genotype-guided treatment strategies in different settings. The implementation of these strategies should be carefully evaluated within specific clinical contexts, considering additional factors such as population characteristics and local costs. Influential variables as the adherence to the use of hearing aids and regular appointments, which were not addressed in this study, could influence the outcomes.

It is also important to highlight that CDDP has numerous indications beyond its use in definitive chemoradiotherapy for HNSCC. The application of risk stratification through genotyping holds potential utility beyond the studied context, including several other tumors in which CDDP is commonly used, such as testicular, breast, gastric, biliary tract and lung carcinomas [63].

## Conclusions

In light of these study findings, we believe future research should focus on prospective clinical trials assessing the implementation of *GSTP1* c.313A>G testing in diverse patient populations. Expanding the genetic panel to include additional known variants of risk for ototoxicity could strengthen the predictive accuracy of pharmacogenetic testing. Given the potential of genetic selection to enhance treatment planning, health policy efforts should also consider the incorporation of pharmacogenetic testing into publicly funded healthcare systems worldwide. Demonstrating the long-term cost savings and improved patient outcomes associated with genotype-guided therapy can support its inclusion in clinical guidelines.

This study highlights the clinical and economic benefits of integrating *GSTP1* c.313A>G genotyping into the management of HNSCC patients treated with CDDP-based chemoradiotherapy. By reducing the incidence of moderate-to-severe ototoxicity, this approach not only improves patient quality of life by diminishing the risk of permanent and debilitating hearing loss but could also generate substantial cost savings for healthcare systems. These results support the importance of pharmacogenomics in advancing precision medicine and optimizing the safety and efficacy of cancer treatments.

## Supporting information

**S1 File. Calculation of probabilities and costs based on the decision analytic model and transitional Markov states.**
(PDF)

**S1 Table. DNA Extraction Costs (in United States Dollars).**
(PDF)

**S2 Table. Real-Time PCR Costs (in United States Dollars).**
(PDF)

**S3 Table. Equipment involved in real-time PCR (in United States Dollars).**
(PDF)

**S4 Table. Manpower costs (in United States Dollars).**
(PDF)

**S5 Table. Costs per PCR test, according to the number of samples evaluated (in United States Dollars).**
(PDF)

**S6 Table. Costs related to treatment (in United States Dollars).**
(PDF)

**S7 Table. Costs related to the use of hearing aids in patients with moderate to severe ototoxicity (in United States Dollars).**
(PDF)

**S8 Table. Cost savings between the conventional treatment group and genotyping, considering the number of simultaneous tests and the volume of patients analyzed.**
(PDF)

**S9 Table. Total costs per patient in the first year for the *GSTP1* c.313A > G genotyping group and the differences compared to the conventional arm, considering simultaneous genotyping of five samples.**
(PDF)

**S10 Table. Median results of *GSTP1* c.313A > G genotyping costs compared to conventional treatment over a ten-year period for a population of 250 patients (in United States Dollars).**
(PDF)

**S11 Table. Inferior limit of credibility results of *GSTP1* c.313A > G genotyping costs compared to conventional treatment over a ten-year period for a population of 250 patients (in United States Dollars).**
(PDF)

**S12 Table. Superior limit of credibility results of *GSTP1* c.313A > G genotyping costs compared to conventional treatment over a ten-year period for a population of 250 patients (in United States Dollars).**
(PDF)

## Author contributions

**Conceptualization:** Ligia Traldi Macedo, Vinicius Eduardo Ferrari, Carmen Silvia Passos Lima.

**Data curation:** Ligia Traldi Macedo, Vinicius Eduardo Ferrari, Ericka Francislaine Dias Costa, Gustavo Jacob Lourenço, Luciane Calonga, Arthur Menino Castilho.

**Formal analysis:** Ligia Traldi Macedo, Vinicius Eduardo Ferrari, Ericka Francislaine Dias Costa.

**Investigation:** Ligia Traldi Macedo, Vinicius Eduardo Ferrari, Ericka Francislaine Dias Costa.

**Methodology:** Ligia Traldi Macedo, Vinicius Eduardo Ferrari, Ericka Francislaine Dias Costa, Gustavo Jacob Lourenço, Arthur Menino Castilho, Carlos Takahiro Chone.

**Project administration:** Ligia Traldi Macedo, Carlos Takahiro Chone, Carmen Silvia Passos Lima.

**Resources:** Vinicius Eduardo Ferrari, Gustavo Jacob Lourenço, Luciane Calonga, Arthur Menino Castilho, Carlos Takahiro Chone.

**Supervision:** Carlos Takahiro Chone, Carmen Silvia Passos Lima.

**Validation:** Vinicius Eduardo Ferrari, Gustavo Jacob Lourenço, Luciane Calonga, Arthur Menino Castilho, Carlos Takahiro Chone, Carmen Silvia Passos Lima.

**Visualization:** Ligia Traldi Macedo, Vinicius Eduardo Ferrari, Ericka Francislaine Dias Costa.

**Writing – original draft:** Ligia Traldi Macedo, Vinicius Eduardo Ferrari.

**Writing – review & editing:** Ligia Traldi Macedo, Vinicius Eduardo Ferrari, Ericka Francislaine Dias Costa, Gustavo Jacob Lourenço, Luciane Calonga, Arthur Menino Castilho, Carlos Takahiro Chone, Carmen Silvia Passos Lima.

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
