## [Decision Letter · Decision Letter 0]

18 Jun 2025

PONE-D-25-07423Pharmacogenetic strategies to mitigate cisplatin-induced ototoxicity in head and neck cancer: a cost-minimization analysis with the use of GSTP1 c.313 A>G genotypingPLOS ONE

Dear Dr. Macedo,

Thank you for submitting your manuscript to PLOS ONE. After careful consideration, we feel that it has merit but does not fully meet PLOS ONE’s publication criteria as it currently stands. Therefore, we invite you to submit a revised version of the manuscript that addresses the points raised during the review process.

We look forward to receiving your revised manuscript.

Kind regards,

Rami M. Elshazli, Ph.D

Academic Editor

PLOS ONE

**Journal Requirements:**

1. When submitting your revision, we need you to address these additional requirements. Please ensure that your manuscript meets PLOS ONE's style requirements, including those for file naming. The PLOS ONE style templates can be found at https://journals.plos.org/plosone/s/file?id=wjVg/PLOSOne_formatting_sample_main_body.pdf and https://journals.plos.org/plosone/s/file?id=ba62/PLOSOne_formatting_sample_title_authors_affiliations.pdf 2. Please note that PLOS ONE has specific guidelines on code sharing for submissions in which author-generated code underpins the findings in the manuscript. In these cases, we expect all author-generated code to be made available without restrictions upon publication of the work. Please review our guidelines at https://journals.plos.org/plosone/s/materials-and-software-sharing#loc-sharing-code and ensure that your code is shared in a way that follows best practice and facilitates reproducibility and reuse. 3. When completing the data availability statement of the submission form, you indicated that you will make your data available on acceptance. We strongly recommend all authors decide on a data sharing plan before acceptance, as the process can be lengthy and hold up publication timelines. Please note that, though access restrictions are acceptable now, your entire data will need to be made freely accessible if your manuscript is accepted for publication. This policy applies to all data except where public deposition would breach compliance with the protocol approved by your research ethics board. If you are unable to adhere to our open data policy, please kindly revise your statement to explain your reasoning and we will seek the editor's input on an exemption. Please be assured that, once you have provided your new statement, the assessment of your exemption will not hold up the peer review process. 4. Please include captions for your Supporting Information files at the end of your manuscript, and update any in-text citations to match accordingly. Please see our Supporting Information guidelines for more information: http://journals.plos.org/plosone/s/supporting-information.

Reviewers' comments:

Reviewer's Responses to Questions

**Comments to the Author**

1. Is the manuscript technically sound, and do the data support the conclusions?

Reviewer #1: Yes

Reviewer #2: Yes

2. Has the statistical analysis been performed appropriately and rigorously?

Reviewer #1: Yes

Reviewer #2: Yes

3. Have the authors made all data underlying the findings in their manuscript fully available?

Reviewer #1: Yes

Reviewer #2: Yes

4. Is the manuscript presented in an intelligible fashion and written in standard English?

Reviewer #1: Yes

Reviewer #2: Yes

5. Review Comments to the Author

**Reviewer #1:** This paper has been written after a thorough analysis after implementation of statistical tools and and a deep analysis of cost minimization has been provided. The author have examined all the areas that needed to be addressed

**Reviewer #2:** This is a well-structured and relevant manuscript that addresses an important clinical and pharmacoeconomic question: whether pharmacogenomic testing (GSTP1 c.313A>G genotyping) can reduce the burden of cisplatin-induced ototoxicity in head and neck squamous cell carcinoma (HNSCC) patients by enabling cost-effective and personalized therapy. The authors used a Bayesian cost-minimization model based on previously published literature, integrating Metropolis-Hastings simulations and Markov modeling to estimate long-term cost savings.

The topic aligns with PLOS ONE’s criteria of sound scientific methodology and meaningful data-driven conclusions. The article offers potential value to clinicians, health economists, and policy-makers.

The methodology is rigorous, using Bayesian inference and probabilistic modeling. The decision tree and Markov models are clearly described and justified. Extensive sensitivity analyses are performed.

Costing is broken down in detail and localized for the Brazilian healthcare system, which adds real-world relevance. The study fills a literature gap in the adult HNSCC population.

The assumption of equivalent oncologic efficacy between cisplatin and docetaxel is critical to the model, yet there is no head-to-head evidence. Authors mention this but should more explicitly discuss its impact on cost-effectiveness outcomes.

Use of pediatric ototoxicity data to estimate some probabilities may reduce generalizability; this should be justified more strongly.

The model doesn’t include Quality-Adjusted Life Years (QALYs), limiting its categorization strictly to cost-minimization; a discussion of this omission would improve clarity.

It is assumed that all patients with hearing loss would receive and adhere to hearing aid treatment over 10 years, which may be optimistic. Addressing adherence variability would strengthen the conclusions.

Writing is generally clear and professional. Figures (decision tree, cost trends) and tables (simulation results, cost breakdowns) are well-organized and support the narrative.

Suggestions for improvement:

The abstract is somewhat long and dense. Consider simplifying language for broader readership.

A graphical summary or flowchart illustrating the pharmacogenomic workflow and outcome paths would benefit visual comprehension.

Define all acronyms upon first use, particularly in the introduction and results (e.g., CDDP, AG/GG, PCR).

Recommendation

Minor Revision

The manuscript is suitable for publication after minor revisions to strengthen its justification of assumptions and improve clarity in some sections.

Required Revisions

Clarify the assumption of oncologic equivalence between cisplatin and docetaxel and how it may influence cost conclusions.

Discuss the potential variability in patient adherence to hearing aid treatment and follow-up, which may impact cost estimations.

Justify the use of pediatric data in adult modeling with more nuance or discuss its limitations more explicitly.

Consider a sentence or two on why QALYs were not modeled and what implications that may have for future research or broader decision-making.

Simplify abstract language and add brief clarifications of acronyms (CDDP, PCR) on first use.

6. PLOS authors have the option to publish the peer review history of their article (what does this mean?). If published, this will include your full peer review and any attached files.

Reviewer #1: **Yes:** Dr. Umar Farooq

Reviewer #2: **Yes:** Nihat Bugra Agaoglu

---

## [Author Response · Author response to Decision Letter 1]

24 Jun 2025

o Reviewer 1:

“This paper has been written after a thorough analysis after implementation of statistical tools and and a deep analysis of cost minimization has been provided. The author have examined all the areas that needed to be addressed.”

Dear reviewer, we sincerely thank you for your thoughtful comment.

o Reviewer 2:

Dear reviewer, we appreciate your suggestions, which certainly helped enhance the clarity of this study . Please find below a detailed description of the revisions made in response to your comments. All changes have also been highlighted in the revised manuscript.

“Clarify the assumption of oncologic equivalence between cisplatin and docetaxel and how it may influence cost conclusions.”

We have included further descriptions on the available evidence of efficacy for docetaxel as a radiosensitizer in the Introduction section, page 4, lines 74-80, as well as Discussion, page 21, lines 394-396. Even though there are no direct head-to-head specific comparisons of cisplatin and docetaxel as single agent radiosensitizers in head and neck squamous cell carcinoma chemoradiation, there is sufficient evidence regarding the use of docetaxel as an alternative agent, and robust results in a randomized trial comparing cisplatin or docetaxel in combination with cetuximab. The assumption of equivalence as a limitation is addressed in the Discussion, page 21, line 384, as in the oroiginal version.

“Discuss the potential variability in patient adherence to hearing aid treatment and follow-up, which may impact cost estimations.”

Great suggestion! In this revised version, we added the variability of adherence rates as a limitation to the interpretation of the results (page 22, lines 417-421).

“Justify the use of pediatric data in adult modeling with more nuance or discuss its limitations more explicitly”

We would like to clarify that the data from the pediatric cohort were not applied as a prior in the model, due to potential differences inherent to that population. To avoid any misunderstanding caused by the previous wording, we have improved the description of the modelling choice in greater detail (page 14, lines 276–282).

“Consider a sentence or two on why QALYs were not modeled and what implications that may have for future research or broader decision-making.”

We added a justification in the Discussion for not including QALYs in this analysis (page 19, lines 344-349). We now acknowledge that including quality-adjusted life years could enhance generalizability and support broader health policy applications in future studies.

“Simplify abstract language and add brief clarifications of acronyms (CDDP, PCR) on first use.”

We revised the Abstract to enhance clarity and readability. Additionally, we reviewed all abbreviations and acronyms to ensure they are properly defined at first use.

We are grateful for the opportunity to improve the manuscript and thank the editorial team and reviewers for their time and guidance.

Yours sincerely,

---

## [Decision Letter · Decision Letter 1]

4 Mar 2026

Pharmacogenetic strategies to mitigate cisplatin-induced ototoxicity in head and neck cancer: a cost-minimization analysis with the use of GSTP1 c.313 A>G genotyping

PONE-D-25-07423R1

Dear Dr. Macedo,

We’re pleased to inform you that your manuscript has been judged scientifically suitable for publication and will be formally accepted for publication once it meets all outstanding technical requirements.

Kind regards,

Rami M. Elshazli, Ph.D

Academic Editor

PLOS One

Additional Editor Comments (optional):

Reviewers' comments:

Reviewer's Responses to Questions

**Comments to the Author**

1. If the authors have adequately addressed your comments raised in a previous round of review and you feel that this manuscript is now acceptable for publication, you may indicate that here to bypass the “Comments to the Author” section, enter your conflict of interest statement in the “Confidential to Editor” section, and submit your "Accept" recommendation.

Reviewer #2: All comments have been addressed

2. Is the manuscript technically sound, and do the data support the conclusions?

Reviewer #2: Yes

3. Has the statistical analysis been performed appropriately and rigorously?

Reviewer #2: Yes

4. Have the authors made all data underlying the findings in their manuscript fully available?

Reviewer #2: Yes

5. Is the manuscript presented in an intelligible fashion and written in standard English?

Reviewer #2: Yes

6. Review Comments to the Author

Reviewer #2: All my concerns have been addressed in this revised manuscript. The revisions significantly improved clarity, methodology, and overall quality. The manuscript is now ready for acceptance.

7. PLOS authors have the option to publish the peer review history of their article (what does this mean?). If published, this will include your full peer review and any attached files.

Reviewer #2: **Yes:** Nihat Bugra Agaoglu

---

## [Editor Report · Acceptance letter]

PONE-D-25-07423R1

PLOS One

Dear Dr. Macedo,

I'm pleased to inform you that your manuscript has been deemed suitable for publication in PLOS One. Congratulations! Your manuscript is now being handed over to our production team.

Kind regards,

on behalf of

Prof. Dr. Rami M. Elshazli

Academic Editor

PLOS One